# Therapeutic effects of adenosine in high flow 21% oxygen aereosol in patients with Covid19-pneumonia

**Pierpaolo Correale[1], Massimo Caracciolo[2], Federico Bilotta[3]\*, Marco Conte[4], Maria Cuzzola[4], Carmela Falcone[5], Carmelo Mangano[6], Antonella Consuelo Falzea[1], Eleonora Iuliano[1], Antonella Morabito[7], Giuseppe Foti[6], Antonio Armentano[8], Michele Caraglia[9,10], Antonino De Lorenzo[11], Michail Sitkovsky[12]\*, Sebastiano Macheda[13]**

1 Medical Oncology Unit, Covid19 Scientific Task Force, Grand Metropolitan Hospital, Reggio Calabria, Italy, 2 Unit of Post Surgery Intensive Therapy (USDO), Covid19 Scientific Task Force, Grand Metropolitan Hospital, Reggio Calabria, Italy, 3 Department of Anesthesiology, Critical Care and Pain Medicine, Policlinico Umberto I, "Sapienza" University of Rome, Rome, Italy, 4 Microbiology Unit, Covid19 Scientific Task Force, Grand Metropolitan Hospital, Reggio Calabria, Italy, 5 Unit of Radiology, Covid19 Scientific Task Force, Grand Metropolitan Hospital, Reggio Calabria, Italy, 6 Unit of Infectious Disease, Covid19 Scientific Task Force, Grand Metropolitan Hospital, Reggio Calabria, Italy, 7 Unit of Pharmacy, Covid19 Scientific Task Force, Grand Metropolitan Hospital, Reggio Calabria, Italy, 8 Unit of Neuro-radiology, Covid19 Scientific Task Force, Grand Metropolitan Hospital, Reggio Calabria, Italy, 9 Department of Precision Medicine, University of Campania "L. Vanvitelli", Naples, Italy, 10 Laboratory of Precision and Molecular Oncology, BiogemScarl, Institute of Genetic Research, Ariano Irpino, Italy, 11 Section of Clinical Nutrition and Nutrigenomic, Department of Biomedicine and Prevention, University of Rome "Tor Vergata", Rome, Italy, 12 New England Inflammation and Tissue Protection Institute, Northeastern University, Boston, Massachusetts, United States of America, 13 Unit of Intensive Therapy and Resuscitation, Covid19 Scientific Task Force, Grand Metropolitan Hospital, Reggio Calabria, Italy

\* federico.bilotta@uniroma1.it, bilotta@tiscali.it (FB); m.sitkovsky@northeastern.edu (MS)

## Abstract

### Background

SARS-Cov2 infection may trigger lung inflammation and acute-respiratory-distress-syndrome (ARDS) that requires active ventilation and may have fatal outcome. Considering the severity of the disease and the lack of active treatments, 14 patients with Covid-19 and severe lung inflammation received inhaled adenosine in the attempt to therapeutically compensate for the oxygen-related loss of the endogenous adenosine→A2A adenosine receptor (A2AR)-mediated mitigation of the lung-destructing inflammatory damage. This off label-treatment was based on preclinical studies in mice with LPS-induced ARDS, where inhaled adenosine/A2AR agonists protected oxygenated lungs from the deadly inflammatory damage. The treatment was allowed, considering that adenosine has several clinical applications.

### Patients and treatment

Fourteen consecutively enrolled patients with Covid19-related interstitial pneumonitis and $PaO_2/FiO_2$ ratio<300 received off-label-treatment with 9 mg inhaled adenosine every 12 hours in the first 24 hours and subsequently, every 24 days for the next 4 days. Fifty-two patients with analogue features and hospitalized between February and April 2020, who did not receive adenosine, were considered as a historical control group. Patients monitoring also included hemodynamic/hematochemical studies, CTscans, and SARS-CoV2-tests.

**Data Availability Statement:** Data underlying the study cannot be made publicly available due to ethical concerns about sensitive patient

information. The data have been deposited on the Grand Metropolitan Hospital database and are available on qualified request (to direzionesanitaria@ospedalerc.it) according to the UE2016/679 GDPR (General Regulation on the protection of sensitive data 2019) law.

**Funding:** The author(s) received no specific funding for this work.

**Competing interests:** The authors have declared that no competing interests exist.

## Results

The treatment was well tolerated with no hemodynamic change and one case of moderate bronchospasm. A significant increase (> 30%) in the $PaO_2/FiO_2$-ratio was reported in 13 out of 14 patients treated with adenosine compared with that observed in 7 out of 52 patients in the control within 15 days. Additionally, we recorded a mean $PaO_2/FiO_2$-ratio increase (215 ± 45 vs. 464 ± 136, P = 0.0002) in patients receiving adenosine and no change in the control group (210±75 vs. 250±85 at 120 hours, P>0.05). A radiological response was demonstrated in 7 patients who received adenosine, while SARS-CoV-2 RNA load rapidly decreased in 13 cases within 7 days while no changes were recorded in the control group within 15 days. There was one Covid-19 related death in the experimental group and 11 in the control group.

## Conclusion

Our short-term analysis suggests the overall safety and beneficial therapeutic effect of inhaled adenosine in patients with Covid-19-inflammatory lung disease suggesting further investigation in controlled clinical trials.

## Background

Covid-19 outbreak has been declared as pandemic by the WHO reporting more than 4 million new cases worldwide with 300,000 related deaths [1]. Almost 20% of these patients developed interstitial pneumonitis that may evolve in ARDS requiring hyperoxic active ventilation, with mostly fatal outcomes [2–5]. The pathogenesis of the Covid-19-related lung injury is still controversial; however, a massive and uncoordinated release of inflammatory cytokines and a post-ischemic reaction to micro-vascular damage and micro-embolization seem to be involved [5–10]. Due to the acute medical need, different drugs are tested in ongoing trials aimed to hamper the effects of the cytokines involved the first phases of the inflammatory process [11–14]. However, mAbs to IL1β (Kanakinumab), IL6-Receptor (Tocilizumab) and inhibitors of Janus kinases (JAK)-1/2 (Baricitinib and Ruxolitinib) did not produce satisfactory clinical outcomes in patients with Covid19-related interstitial pneumonitis [14–16]. The most recent clinical evidences highlight a 20–30% mortality rate in patients with Covid19-related lung injury requiring active oxygen ventilation and depending on the classification of patients, time of intervention and how critical their illness is [17–19]. These reports are leading to the suspicions that some iatrogenic complication other than a mechanical lung damage does occur [17–19]. Accordingly, we carried out a clinical investigation based on insights and therapeutic suggestions offered in preclinical studies and paper with the self-explanatory title: "Oxygenation inhibits the physiological tissue-protecting mechanism and thereby exacerbates acute inflammatory lung injury" [20]. Based on the results of those preclinical studies in mice we hypothesized that mechanical ventilation and hyper-oxygenation lead to the unacceptable inflammatory side effects in patients with Covid19-related severe pulmonary complications. We further assumed, that the otherwise life-saving oxygenation also weakens the local tissue hypoxia-driven and adenosine A2A receptor (A2AR)-mediated anti-inflammatory mechanism [21–23]. Without this major physiological anti-inflammatory lung tissue protection, the neutrophils, lung macrophages and pulmonary natural killer cells in lungs are no longer inhibited and are unleashed to destroy the still healthy lung [20, 24–26]. Our clinical investigation was

also enabled by long-term studies of the role of A2A or A2B adenosine receptors in inflammation and in rheumatology [26, 27]. We reasoned that pharmacologically compensating for the oxygenation-associated loss of the naturally generated extracellular adenosine in inflamed lungs of COVID-19 patients would represent a possible solution of the explained above pathophysiological dilemma in oxygenating the COVID-19 patients. In support of such intervention, preclinical data demonstrate that the deadly immunological side effects of the supplemental oxygenation are prevented by the intra-tracheal injection of an adenosine analog, a synthetic A2AR agonist, to compensate for the oxygenation-related loss of the lung tissue-protecting adenosine [27–30]. Having this solid scientific rationale and in absence of effective therapies for patients with COVID-19 and severe interstitial pneumonitis, it was decided to use the inhalatory adenosine as an anti-inflammatory drug to compensate for the oxygen-related loss of the naturally generated nucleoside adenosine in inflamed lungs. Due to the acute medical need and shortage of time, it was impossible to gain access to a synthetic A2A receptor agonists, while adenosine was already available for clinical use (Adenoscan® and Krenosin®) with a number of different applications in humans [31]. Additionally, adenosine has already been tested as an aerosol formulation (6 to 40 mg per dose), presenting an acceptable safety profile with no hemodynamic or other side-effects in normal subjects, and it is currently used to discriminate patients with small respiratory tract asthmatic disease from chronic inflammatory lung disease [32–35]. In this retrospective data analysis, we report on safety and efficacy of adenosine as inhalatory formulation to patients with severe Covid-19-realted interstitial pneumonitis with a $PaO_2/FiO_2$ratio <300 requiring ventilator supports as a compassionate life-saving therapeutic act.

## Methods

### Patient population and treatment

Fourteen hospitalized patients with a $PaO_2/FiO_2$ratio <250, a radiological picture suggestive of severe interstitial pneumonitis, positive for the expression of SARS-Cov-2 who resulted unresponsive to previous treatments with hydroxychloroquine (HC), azitromycin (AZM) and low molecular weight heparin (LMWH)or corticosteroids signed an informed consent and received an off-label treatment with inhalatory adenosine at the dosage of 9 mg every 12 hours in the first 24 hours and subsequently, every 24 days for four consecutive days. Adenosine was nebulized and dispensed by an Aerogen USB Controller linked to a high flux device with 21% $FiO_2$, a flow of 60 l/m in five minutes. Inhaled adenosine dose was extrapolated from the preclinical studies in mice [25, 31–33] as well as from the clinical studies using adenosine as aereosol formulation showing dose limiting efficacy over 10 mg and no adverse events in normal individuals and patients with non asthmatic disease [31–35]. No concomitant treatment for Covid19 including corticosteroids was allowed. The off label treatment and patient monitoring was approved for each single individual by the Hospital Safety Team and by the Ethical Committee of South Calabria (April, 30th, 2020). Patients' privacy and sensitive data were appropriately protected and database was available on the Grand Metropolitan Hospital database available on appropriate request (direzionesanitaria@ospedalerc.it) according to the UE2016/679 GDPR (General Regulation on the protection of sensitive data 2019) law, published on May 25th, 2018 on the "GazzettaUfficale Repubblica Italiana".

A copy of the treatment protocol has been deposited to enhance the reproducibility of our results at http://journals.plos.org/plosone/s/submission-guidelines#loc-laboratory-protocols. "A rapid assay for the detection of SARS-CoV-2 (COVID-19) was used: Seegene 'Allplex™ 2019-CoV Assay, catalognumber #RP10243X 100 rxn, targeting SARS-CoV-2 RdRp, E and N genes. It was approved for emergency use by the U.S. Food and Drug Administration (FDA),

Health Canada and Korea Centers for Disease Control and Prevention (KCDCP) and also by the CE-IVD marked. Assays were performed by means of a Seegene's automation platform allows a unique streamlined work flow for detection of COVID-19. Equipments included NIMBUS IVD instrument for automated extraction and PCR setup, CFX96™for Real-time PCR and Seegene Viewer for automated data analysis. Initial data analysis was performed in the CFX96 Manager software prior, followed by export to the Seegene viewer software. The SARS-CoV-2 results were automatically categorised by the 'Seegene viewer' software as '2019-nCoV detected', 'negative' or 'invalid' based on predefined parameters (Allplex 2019-nCoV assay IFU) [36].

## Statistical considerations

Even though the use of aereosolized Adenosine was performed on compassionate off-label use, the successful preliminary results led us to design a controlled trial presently submitted for approval to the Italian Drug Authority (AIFA) **EudraCT:** 2020-002007-19, code ADS-PNM-1) and National Covid19 Ethical Committee. We hypothesized that the above mentioned treatment deserved further consideration only if it resulted active in more that 70% of the patients within 2weeks. For the instance, patients were considered as treatment responsive if they showed a 30% increase in the $PaO_2/FiO_2$ratio within 15days. This endpoint was extrapolated by the results of the literature aimed to investigate Covid19 specific treatments [6] and from our historical control including 52 hospitalized patients with Covid19 pneumonitis presenting analogue clinical conditions as those who received Adenosine. The patients in the control group received the most conventional treatments for Covid19 and did not show a significant treatment-response in more than 30% of the cases within 15days. Thus, assuming an alpha and a beta error of 5% and 20% respectively, a total of 28 responses out of 40 patients (vs. 16 responses out of 40 cases in the control group) should be enrolled to achieve a statistically significant difference ($P \leq 0.05$). On these bases, our adenosine treatment should have been considered as completely inactive in case of less than 10 recorded responses within the first 14consecutively enrolled patients (71%).

The treatment should have been also discharged in the case of grade 3–4 adverse events (World Health Organization toxicity scale) occurring in at least 3cases within the first 10consecutively treated patients.

## Ancillary study

Patients were daily monitored for vital parameters, arterial pressure (AP) and heart-rate (HR), hemogas-analysis, blood cell counts, biochemistry, ECG, inflammatory markers (CRP, LDH and ESR), coagulation asset and D-Dimer. Interleukin-6 was also measured inthe serum of ten out 14 patients at baseline and 120 hours after the beginning of the treatment by Electroluminescent immune Assay (kit-ECLIA, Roche) in the laboratory of Clinical Pathology of the Grand Metropolitan hospital (GOM), RC, Italy.

The detections of SARS-Cov-2 in upper or lower upper or lower respiratory specimens was performed at baseline,48, 120 hours and 15 days after the beginning of treatment. It was evaluated by laboratory of Microbiology& Virology of the GOM, RC, Italy. Upper (nasopharyngeal swabs) and lower (broncho-alveolar lavages, broncho-aspirates and tracheal aspirates) respiratory tract specimens, were collected using Copan Universal Transport Medium (UTM-RT®) System or sterile container at 4˚C and processed within 24 hours. Real-time reverse transcription-PCR is currently the most reliable diagnostic method for COVID-19 around the world. RNA-COVID 19 was evaluated by using an Allplex 2019-nCoV Assaythat identifies three different target genes: E (envelope), RdRp (RNA-dependent RNA polymerase, and N (nucleoprotein gene) according to

the international recommended guidelines by the World Health Organization. This test has also received CE-IVD mark and KFDA approval. The test assay was performed following the manufacturer's instructions using with the appropriate equipment (http://www.seegene.com/assays/allplex_2019_ncov_assay). According to the interpretation criteria, detection of only one of multiple genes has been interpreted as COVID-19 positive.

Thorax X rays and/or High resolution CT scan were performed at baseline and after treatment and results were analysed by the same dedicated radiologists.

## Statistical analysis

The between-mean differences were statistically analyzed using Stat View statistical software (Abacus Concepts, Berkeley, CA). The results were expressed as the mean±S.D. and the differences determined using the two-tail Student's *t*-test for paired samples. A *P*-value of 0.05 or less was considered statistically significant.

## Results

### Patients population

Fourteen consecutively enrolled patients, 10 males and 4 females with a mean age of 57± 19 years were approved to receive life-saving off-label treatment with inhaled adenosine, dispensed by a high flow device and 21% $O_2$ starting on April 10[th] 2020. All these patients were positive for the expression of SARS-Cov-2 and had been previously hospitalized. Thirteen of them had received empiric treatments commonly used for Covid19 with no clinical or biological improvement for more than 3 weeks. Before receiving adenosine they had presented a $PaO_2/FiO_2$ratio <250 and a CT scan picture suggestive of severe interstitial pneumonitis. Eleven patients had received previous treatment with HC, AZM, LMWH. Four of them had also received previous off-label treatments with Tocilizumab more than 3 weeks prior adenosine administration. Ten patients, hosted in the Unit of Infectious Diseases required high flux oxygenation, while further4 patients requiring active ventilation, were hosted in the Resuscitation unit of the Grand Metropolitan Hospital (Table 1). Fifty-two patients with similar clinical/radiological features were hospitalized in the Grand Metropolitan Hospital starting on March 10[th], 2020 and were considered as the historical control of the study. Forty-nine of the latter patients, who required oxygenation with Ventimask (41 cases) or CPAP with helmet (8 cases), were hosted in the Infectious Diseases/Covid19Operative Unit while 3 further patients requiring mechanic ventilation, were hosted in the Covid19 Resuscitation Unit [36]. The latter patients had received specific treatment with HC, AZM, LMWH (40 cases) and Tocilizumab (12 cases) and none of them received adenosine.

### Adverse events

Adenosine treatment was well tolerated and there was no effect on either mean HR (75±7 vs. 75±10bpm at 24 hours, P>0.05) or mean AP [118 (±15)/75 (±10) vs. 121 (±10)/77 (±8) mmHg at 24 hours]during and after the treatment procedure. There was a case of reversible bronchospasm during the third adenosine dose administration in a mechanically ventilated patient that consequently discontinued the treatment (Table 1). A temporary flushing was also recorded in another case 6 hours after the first adenosine dose.

### Treatment response

Inhalatory adenosine administration resulted in a 30%-rise in the $PaO_2/FiO_2$ratio since the beginning of the treatment in 13 out of 14 consecutively treated patients (93%) within 15 days

**Table 1. Patient features, treatments, and outcome.**

| Pts code | Co-morbidities and Previous treatments | Baseline Performance status and respiratory needs (score 1–4) | CoVid RNA expression | Clinical benefit (score 0–5)/ Radiological response (Score 0–5)/ Adverse events (g WHO score 1–4) |
|---|---|---|---|---|
| -1- | Right bundle branch block, Keratoconus | ECOG 2 | baseline: Pos | Good (4) /ND/No AEs |
| | HC + AZR, LMWH 4,000U/bd | Ventimask | 48 h: Neg | |
| | | | 120 h: Neg | |
| | | | 15 days: Neg | |
| -2- | Atopic allergy, Hypertension | ECOG 2 | baseline: Pos | Excellent (5)/4/No AEs |
| | HC + AZR, LMWH 4,000 U/ bd | Ventimask | 48 h: Low Exp | |
| | | | 120h: Pos N gen | |
| | | | 15 days: Neg | |
| -3- | None | ECOG 2 | baseline: Pos | Excellent (5)/3/No AEs |
| | HC + AZR, LMWH 6,000 U/ bd | Ventimask | 48h: Low exp | |
| | | | 120h: Pos N gene | |
| | | | 15 days: Neg | |
| -4- | Mitral insufficiency, atrial fibrillation, hypertension | ECOG 2 | baseline: Pos | Excellent (5)/4/No AEs |
| | HC + AZR, LMWH 6,000 U/bd | Ventimask | 48h: Low exp | |
| | | | 120h: Neg | |
| -5- | None | ECOG 2 | baseline: Pos | Pos (5) /4/(g1 –Flushing) |
| | HC + AZR, LMWH 4,000 U/ bd | Ventimask | 48h: Low exp | |
| | | | 120h: Pos N gene | |
| | | | 15 days: Neg | |
| -6- | RCU, Iatrogenic hypothyroidism, Obesity | ECOG 3 | Baseline: Pos | Good (4) /4/No AEs |
| | Tocilizumab, HC + AZR, LMWH 4,000 U/bd | CPAP with helmet | 48h: Neg | |
| | | | 120h: Neg | |
| -7- | Psychiatric disease | ECOG 2 | baseline: Pos | Excellent (5) /ND/No AEs |
| | HC + AZR, LMWH 4,000U/bd | Ventimask | 48h: Pos | |
| | | | 120h: Neg | |
| | | | 15 days: Neg | |
| -8- | Atopic allergy, asthma, mitral insufficiency | ECOG 2 | baseline: Pos | Excellent (5) /ND/No AEs |
| | None | Ventimask | 48h: Pos | |
| | | | 120h: Pos N gene | |
| | | | 15 days: Neg | |
| -9- | None, Bacterial Pneumonia | ECOG 2 | baseline: Pos | Excellent (5) /1 –pre-existing bacterial pneumonia /No AEs |
| | LMWH 4,000 U/ bd | Ventimask | 48h: Neg | |
| | | | 120h:Neg | |
| | | | 15 days: Neg | |
| -10- | None | ECOG 2 | baseline: Pos | Good (4) /3/(g2 –Nausea) |
| | HC + AZR | Ventimask | 48h: Pos N gene | |
| | | | 120h: Neg | |
| | | | 15 days: Neg | |
| -11- | Hypertension, Prostate hypertrophy | ECOG 4 | baseline: Pos | Moderate (3) /1/No AEs |
| | Tocilizumab, HC + AZR, LMWH 6,000 U/bd | IOT | 48h: Pos N gene | |
| | | VM, Tracheotomy | 120h: Neg | |
| | | | 15 days: Neg | |

(*Continued*)

**Table 1.** (Continued)

| Pts code | Co-morbidities and Previous treatments | Baseline Performance status and respiratory needs (score 1–4) | CoVid RNA expression | Clinical benefit (score 0–5)/ Radiological response (Score 0–5)/ Adverse events (g WHO score 1–4) |
|---|---|---|---|---|
| -12- | Alzheimer Disease, bladder cancer Osteoporosis, Urinary tract infections | ECOG 4 | baseline: Pos | Good (4) /2/No AEs |
| | | NIV | 48h: Pos N gene | |
| | Ritonavir, HC + AZR | CPAP with helmet | 120h: Neg | |
| | | HFNC | 15 days: Neg | |
| -13- | Tongue carver, Hypertension, COPD | ECOG 4 | baseline: Pos | Good (5) /ND/No AEs |
| | Tocilizumab AZT, LMWH 6000 U/bd | CPAP with helmet | 48 h: Pos N gene | |
| | | | 120 h: Pos N gene | |
| | | HNFC | 15 days: Neg | |
| -14- | Hypertension, Prostate Hyperplasia | ECOG 4 | baseline: Pos | Poor (2) /2/(g3-Bronchospasm) |
| | Ritonavir, Tocilizumab HC + AZR, LMWH 6000 U/ bd | CPAP with helmet | 48h: Pos | |
| | | | 120h: Pos N gene | |
| | | IOT, VM | 15 days: Pos | |

Patient performance status at baseline was evaluated according to the Eastern Cooperative Oncology Group (ECOG) scale (1–4); adverse events (AE) were evaluated according to the World Health Organization (WHO) scale grade (g).

HC = hydroxychloroquine, AZR = Azitromycin, IOT = oro-tracheal intubation, VM = Mechanical ventilation; CPAP = Continuous positive airway pressure; HNFC = High Flow oxygenation; COPD = Chronic Obstructive Pulmonary disease.Low molecular weight heparins = LMWH; Bidaily = bd; CT score finding: -1 = worse; -2 = no change; -3 = slight improvement (reduction focal or diffuse pneumonia <50%); -4 = improvement > 50%; -5 = no evidence; -ND = not done.

from the beginning of the treatment, thus fulfilling the statistically endpoint of this preliminary analysis that was pre-fixed at 72%. In particular, the average $PaO_2/FiO_2$ ratio showed a significant increase from 215 ± 45 to 464 ± 136, P = 0.0002 in 120 hours (Fig 1A) with a median time

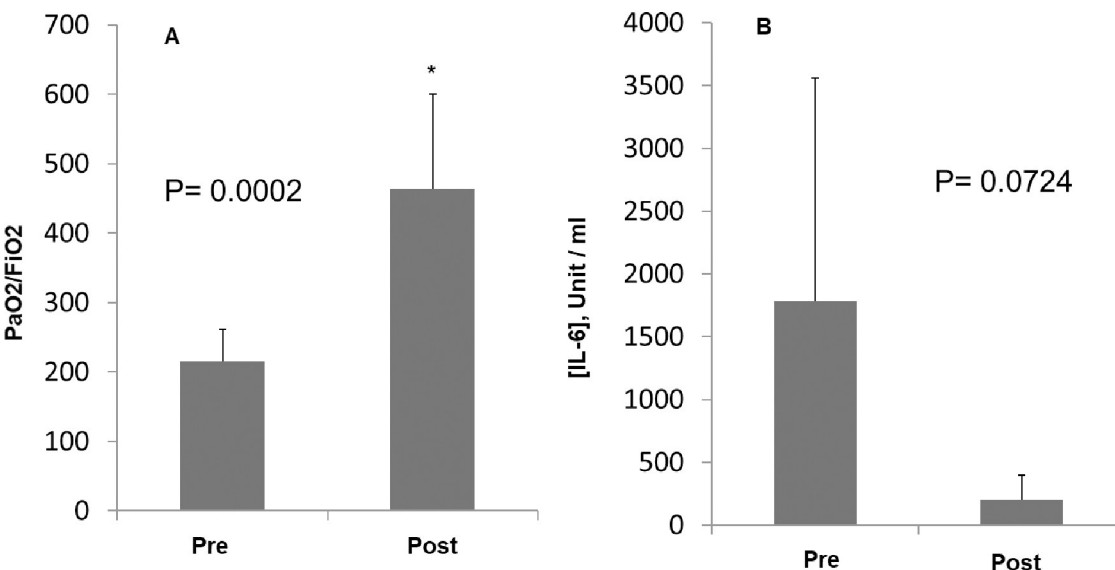

**Fig 1. Respiratory and inflammatory marker monitoring before and after adenosine treatment.** Panel A)- Adenosine treatment shows a significant improvement in the mean $PaO_2/FiO_2$ratio in 14 patients who received adenosine. Panel B)- The plot shows a post-treatment decline in IL-6 serum level. However, the differences did not achieve statistical significance (P = 0.07). There was no significant treatment-related changes in blood cell counts as well as serum C-reactive protein and Lactate Dehydrogenase (LDH) levels.

to full recovery of 6 ± 2 days. On the other hand, in the control group of patients who received standard treatments with HC, AZM, LMWH and Tocilizumab (12 cases only), there was a30% increase in $PaO_2/FiO_2$ratio within 15 days only in 7out of52 patients (13.5%) with no significant rise after 120 hours (210±75 vs. 250±85 P>0.05) and a median time to full recovery of 21 ± 5.5 days.

As a further consideration, 13 patients who received adenosine treatment presented a clinical benefit with decrease in symptoms and improvement in performance status within 3days. Eight of them did not require further oxygen administration and could be released by the hospital within 1 week from the beginning of the treatment. Two patients in active ventilation were extubated 72 hours after the beginning of the treatment and addressed to high flux ventilation out of the resuscitation facility. A high resolution CT scan monitoring was performed in only 10 patients as the remaining4 patients refused any post treatment scan. A complete resolution of the lung disease was recorded in 2 patients and a significant improvement of the picture was observed in additional4 cases (Fig 2). Two patients showed a minimal radiological benefit, while 2 patients showed the presence of new lung consolidative areas and pleural effusion suggestive of a new bacterial complication.

Finally, there was one Covid-19 related death among the 14 patients (7.14%) in the experimental group and 11 out of 52 cases (21.11%) in the control group. The latter group was in the average range of lethality reported for homologous Covid-19 hospitalized patients worldwide.

## Laboratory response

Our analysis revealed some limited evidence for serum decline of IL-6 levels120 hours after the end of the treatment that, however, did not achieve statistical significance (P = 0.075) (Fig 1B). The detection of SARS-Cov-2 was performed at baseline and 48 and 120 hours and 15 days after the beginning of adenosine treatment. Eight/14 patients showed complete disappearance of viral load while 5 of them showed the persistence of a very low virus load. In the latter patients, only SARS-Cov-2 N gene could be detected, at higher Ct respect to baseline value which disappeared above the 40 Ct. As an additional finding, we observed early decrease in the virus load with no detection in RdRp/E genes already at 48 hours after the beginning of the treatment (Fig 3 and Table 1). All of the 13 responsive patients resulted SARS-Cov-2 free after 15 days (Fig 3) and no case of reinfection was recorded at 4 months after the end of the treatment. Among the 52 patients in the control group only 5 patients showed SARS-Cov-2 expression decline and disappearance within 15 days of other conventional treatments.

## Discussion

Here we provide a retrospective analysis of 14 hospitalized patients with Covid19-related inflammatory lung injury who received nebulised adenosine. Our results showed no adverse events and a treatment response rate, considered as an at least30%-increase in $PaO_2/FiO_2$ratio at 15 days, in 13 out of14 adenosine-treated patients (92%). The response rate was much higher than that reported in the analogue52control patients who only showed a treatment response rate of only 13.5% at 15 days. On these bases, our adenosine treatment fulfilled the prefixed statistical endpoint of activity of 70% in the first 14 consecutively treated patients and could be considered for further studies. The clinical results of the treatment were very promising considering that patients' symptoms (respiratory as well as fever, asthenia, headache) declined within 4 days from the beginning of the treatment. Eight of the adenosine-treated patients could be released from the hospital within one from since the beginning of the treatment with adenosine. These clinical results were also supported by radiological imaging study whose results showed a significant improvement in the signs of interstitial lung pneumonitis in 6 out

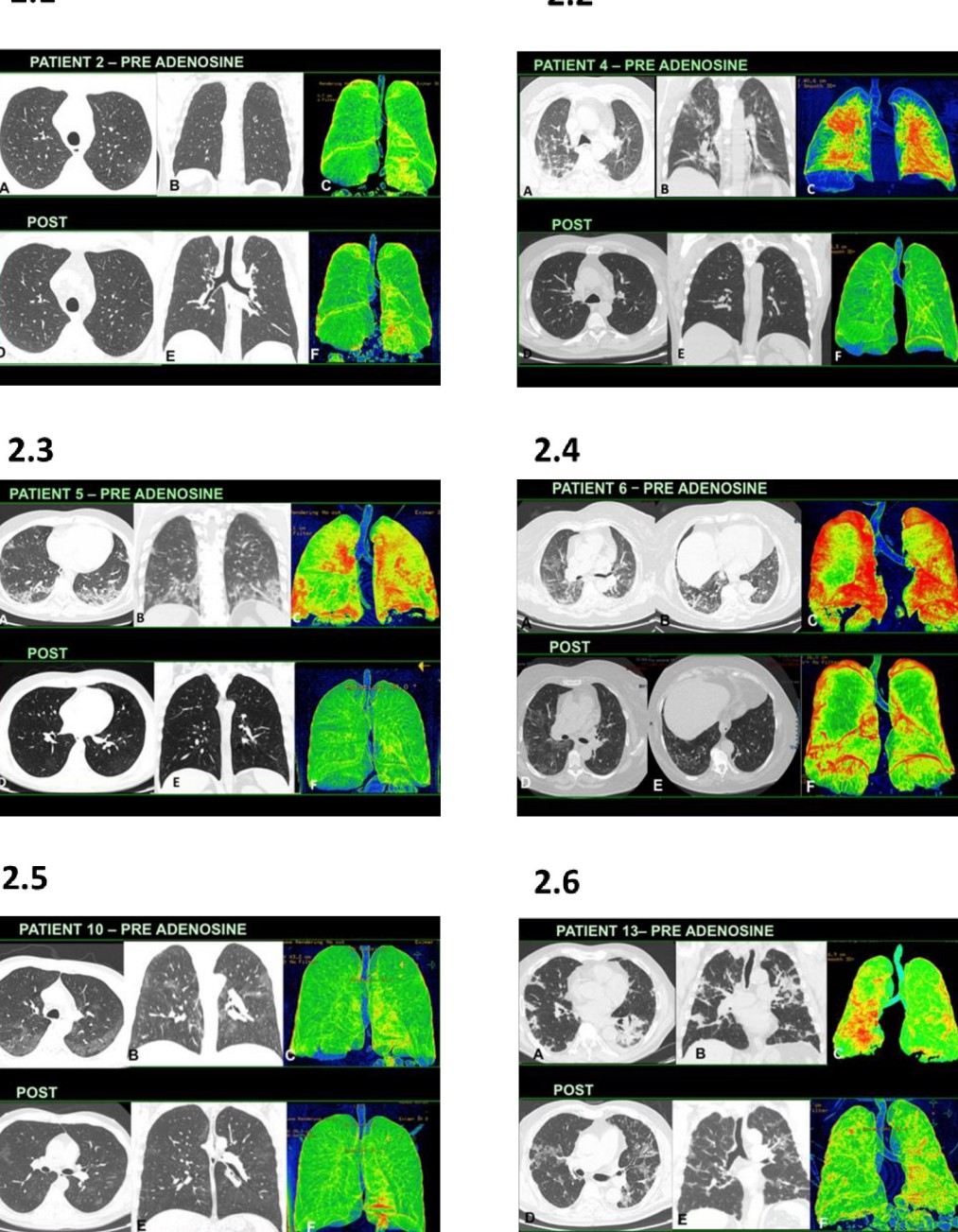

**Fig 2. The High Resolution Computerized Scan (HRCT) monitoring before and after adenosine treatment.** Panel. 2.1– Patient #2- (A-B) baseline HRCT shows signs of interstitial pneumonitis with focal area of ground-glass in the RSL; (C) §Pre-treatment volume rendering. (D-E) Post-treatment HRCT shows widespread reduction in the interstitial pneumonitis. (F) §Post-treatment volume rendering. Panel 2.2 -Patient #4- (A-B) baseline HRCT shows signs of interstitial pneumonitis with areas of ground-glass in the RSL and LIL with pleura-parenchymal branches in the periphery; (C) §Pre-treatment Volume rendering. (D-E) Post-treatment HRCT shows wide-spread reduction in the interstitial engagement.(F) §Post-treatment volume rendering shows reduction in amorphous increase in lung density. Panel 2.3 -Patient #5- (A-B) baseline HRCT shows widespread signs of interstitial pneumonitis with areas of ground-glass and pleura-parenchymal branches, present in both lungs with spread to the periphery in the LIL and RIL.(C) §Pretreatment volume rendering. (D-E) post-treatment HRCT shows widespread reduction in the interstitial engagement. (F) §Post-treatment volume rendering shows a significant reduction in the parenchymal thickenings. Panel 2.4-Patient #6- (A-B) baseline HRCT shows widespread signs of interstitial pneumonitis with areas of groundglass and crazy paving, present in both lung fields with spreading to the

periphery.(C) §Pre-treatment volume rendering. (D-E) Post-treatment HRCT shows widespread reduction in interstitial engagement and crazy paving. (F) §Post-treatment volume rendering. Panel 2.5-Patient#10-(A-B) baseline HRCT shows widespread signs of interstitial pneumonitis with areas of groundglass in both lungs with spreading to the periphery. (C) §Pretreatment volume rendering. (D-E) Post-treatment HRCT shows widespread reduction in interstitial engagement. (F) §Post-treatment volume rendering. Panel 2.6-Patient#13- (A-B) Baseline HRCT shows widespread signs of interstitial pneumonia, fibrous septa and pulmonary thickening, with areas of groundglass in both lungs spreading to the periphery. (C) §Pretreatment volume rendering. (D-E) Post-treatment HRCT shows widespread reduction in interstitial pneumonia and septal thickening. (F) §Post-treatment volume rendering showing reduction thickenings. §In the volume rendering study, green area represent the normal lung parenchyma while red areas indicate the inflammatory involvement.

of 10 studied patients. As an additional and unexpected finding we also detected a significant decrease in SARS-Cov-2 viral load. These results were not observed in our control group of patients where $PaO_2/FiO_2$ ratio and clinic-radiological improvement as well as SARS-Cov-2 disappearance did not occur in less than 30 days. However, limitations of the present study are both the small patient sample size and the historical nature of the comparison; despite these facts, the death rate in the group of patients receiving adenosine was 7.14% (1/14 patient), while those recorded in the group of patients who did not receive adenosine was 21.11% (11/52 patients). The results observed in our historical group were, therefore, in line with those reported worldwide for similar patients [6]. Our findings seem to confirm the results of the preclinical studies in mice were intra-bronchial administration of A2 receptor agonists could restore the anti-inflammatory and tissue protective effects of oxygen-dampened adenosine system [20, 32–35] (S1 Fig). Presently, we do not have enough information to explain the effect of inhaled adenosine therapy on SARS-Cov-2 load; however, it can be hypothesized that adenosine given in hypoxic conditions is able to induce the following effects: i) to restore an appropriate virus-specific immune-response previously attenuated by the inflammatory storm; ii) to exert a direct anti-viral host effect mediated throughout the A2R pathway; iii) to convert the intracellular adenosine in pro-apoptotic metabolites (like deoxy-adenosine/deoxy-ATP) in some infected cells. The latter effect should be investigated in future studies. In this light, there is large concordance on the fact that mechanical damage to the lung associated to active ventilation can add to the SARS-CoV-2 -induced lung damage [5]. On the other hand, it is now widely known that abuse of hyperoxic breathing itself can inhibit the major physiological tissue-protecting hypoxia-A2-adenosinergic mechanism leading to massive tissue damage consequences. Our treatment was aimed to restore the A2 adenosine receptors signaling and thereby ensuring again the protection of healthy lung tissue–even in the presence of continuing oxygenation.

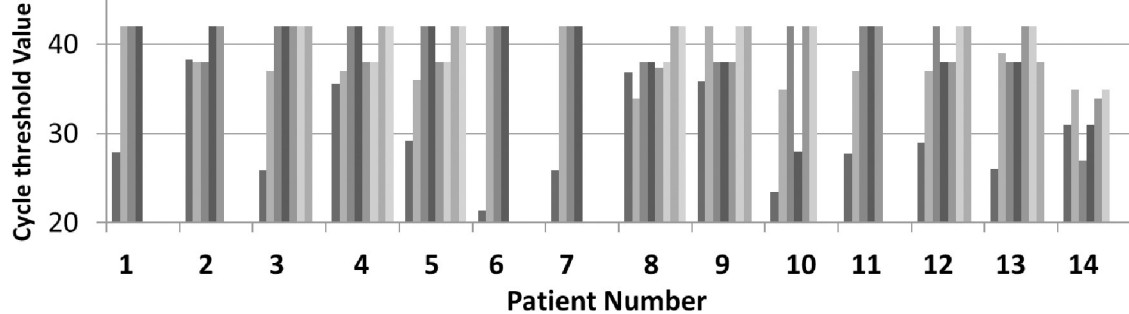

**Fig 3. The monitoring of threeSARS-CoV-2 RNA target genes before and after adenosine treatment.** Evaluation of SARS-COV-2 RNA N gene Ct value in respiratory specimens over time. N gene Ct value peaked at the baseline and decreased in responder patients. Above 40 Ct RNA-N genes were considered not detectable. From the left to the right: 1st column, baseline; 2nd column, 48 hours; 3rd column, 120h; up to 15 days (8th column) from the beginning of the treatment.

In agreement with the previous preclinical studies, we showed that our treatment strategy in this small patients' series, could result in an accelerated increase in $PaO_2/FiO_2$ ratio and performance status and an antiviral effect. In this view, inhaled adenosine is the first treatment for Covid-19 aimed to exert rapidly both clinical benefit and antiviral activity in critical patients. We believe that our results definitely deserve to be investigated in controlled clinical trials supported by both clinical as well as an immune-biological monitoring and expertise. Considering the dramatic consequence that Covid-19 outbreak is determining worldwide, these results, if confirmed in a complete clinical trial assessment, could greatly improve how we treat and cure these patients.

## Supporting information

**S1 Fig.**
(DOCX)

## Acknowledgments

We wish to thank all of the patients and their families for trusting us in allowing the off-label treatment. We wish to thank all of the paramedic personnel, Biologists and Technicians within the Grand Metropolitan Hospital-Covid-19 Task Force for their dedication to patients' care and monitoring. A special mention for their precious work to Drs. Rosa Basile, Maria Stella Carpentieri, Giuseppe Ieropoli, Alfredo Kunkar, Maria Polimeni, Domenico Sofo, Saverio De Lorenzo, Infectious Disease Unit; Drs Nicola Arcadi, Anna Ferrarelli, Pietro Arciello, Andrea Sergi, Radiology Unit; Dr. Antonella Meliadò, MicrobiologyUnit; Dr Rocco Giannicola, Medical Oncology Unit; Drs Maria Altomonte e Antonio Nesci, PharmacyUnit; Drs. Marco Tescione, Demetrio Labate, Stefano La Scala, Giuseppe Martino, Francesco Curmaci, Eugenio Vadalà, Rosalba Squillaci, Graziella Marano, Nicola Polimeni, Enzo Battaglia, Giuseppe Sera, Caterina Morabito, Intensive therapy and Resuscitation Unit, Covid19 Scientific Task Force, Grand Metropolitan Hospital, Reggio Calabria, Italy.

We wish a particular acknowledgment to the Extra-ordinary General Manager of the Grand Metropolitan Hospital of Reggio Calabria, Ing. Iole Fantozzi and her strategic team that supported the study and was able to organize in a few weeks an enthusiastic and collaborative multidisciplinary task force to fight against Covid19.

## Author Contributions

**Writing – original draft:** Pierpaolo Correale, Massimo Caracciolo, Federico Bilotta, Marco Conte, Carmela Falcone, Carmelo Mangano, Antonella Consuelo Falzea, Eleonora Iuliano, Antonella Morabito, Giuseppe Foti, Antonio Armentano, Michele Caraglia, Antonino De Lorenzo, Michail Sitkovsky, Sebastiano Macheda.

**Writing – review & editing:** Pierpaolo Correale, Massimo Caracciolo, Federico Bilotta, Marco Conte, Maria Cuzzola, Carmela Falcone, Carmelo Mangano, Antonella Consuelo Falzea, Eleonora Iuliano, Antonella Morabito, Giuseppe Foti, Antonio Armentano, Michele Caraglia, Antonino De Lorenzo, Michail Sitkovsky, Sebastiano Macheda.

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
