## [Decision Letter · Decision Letter 0]

30 Jul 2020

PONE-D-20-18007

Aerosolized adenosine for the treatment of ICU Covid-19 patients.

PLOS ONE

Dear Dr. Bilotta,

Thank you for submitting your manuscript to PLOS ONE. After careful consideration, we feel that it has merit but does not fully meet PLOS ONE’s publication criteria as it currently stands. Therefore, we invite you to submit a revised version of the manuscript that addresses the points raised during the review process.

ACADEMIC EDITOR: I have received the comments of the reviewers on your manuscript. The specific comments of the reviewers are included below. Please provide point by point response in your revised manuscript.

We look forward to receiving your revised manuscript.

Kind regards,

Muhammad Adrish

Academic Editor

PLOS ONE

Journal Requirements:

2. Thank you for stating in the text of your manuscript that patients signed informed consent and that "use of the data for retrospective study was approved in the Calabria South ethics committee". Please also add this information to your ethics statement in the online submission form.

3. Please provide the catalog number for the Allplex 2019-nCoV Assay in your methods. In addition, please ensure that you describe the sources and models of your equipment in the methods section of your manuscript.

4.We note that you have indicated that data from this study are available upon request. PLOS only allows data to be available upon request if there are legal or ethical restrictions on sharing data publicly. For information on unacceptable data access restrictions, please see http://journals.plos.org/plosone/s/data-availability#loc-unacceptable-data-access-restrictions.

6. Please include a copy of Table 1 which you refer to in your text on page 19.

7. Please include a caption for "Figure additional matherial.pptx".

8. Please ensure that you refer to Figure "Figure additional matherial.pptx" in your text as, if accepted, production will need this reference to link the reader to the figure.

Reviewers' comments:

Reviewer's Responses to Questions

**Comments to the Author**

1. Is the manuscript technically sound, and do the data support the conclusions?

Reviewer #1: Yes

Reviewer #2: Yes

Reviewer #3: No

Reviewer #4: Yes

2. Has the statistical analysis been performed appropriately and rigorously? 

Reviewer #1: Yes

Reviewer #2: Yes

Reviewer #3: No

Reviewer #4: I Don't Know

3. Have the authors made all data underlying the findings in their manuscript fully available?

Reviewer #1: Yes

Reviewer #2: Yes

Reviewer #3: No

Reviewer #4: No

4. Is the manuscript presented in an intelligible fashion and written in standard English?

Reviewer #1: Yes

Reviewer #2: Yes

Reviewer #3: No

Reviewer #4: Yes

5. Review Comments to the Author

Reviewer #1: To improve the treatment of severe covid-19 is very important and the study is thus highly relevant.

This early study without control group show more that the treatment is well tolerated, the effect is not proven without controls.

In the Background it is written at row 4 that intersitial pneumonitis due to covid -19 is mostly fatal, which is no longer true and at row 13 it is written that 65-80 % of the ventilator treated patients die, which is too high in most recent clinical materials, at several centers now around 20%. It is dependent on how the patients are selected and how critically ill they are.

The recently shown positive effect of high dose corticosteroids is important to control for in a future prospective trial on Aerosolized adenosine. Is Aerosolized adenosine adding an effect on case fatality and duration of hospital stay in addition to the effect of high dose corticosteriods ?

Reviewer #2: Aerosolized adenosine can be effective in patients with advanced lung damage due to covid19 and needing mechanical ventilation. It is necessary to perform controlled studies on whether adenosine is effective in these types of patients. This study may be a guide for randomized controlled trials.

Reviewer #3: This manuscript contains an attempt to test whether inhaled adenosine can be used as treatment for COVID-19 patients that require oxygen ventilation. The authors decided to test the anti-inflammatory effect of adenosine to treat COVID-19 patients based on previous results from a knockout mice animal model that showed an immunosuppression of Adenosine receptor (A2AR) associated to oxygen ventilation, which was improved through the use of the Adenosine agonist CGS21680.

The authors selected 14 patients that were previously treated with other experimental treatments, such as hydroxychloroquine, azytromycin, low molecular weight heparin, and tolicizumab (it’s not clear whether the other treatments were performed during the experiment period), to participate on the experiment. The selection criteria were not clearly defined and there was no control group. The manuscritp comparison was restricted to analyzing the patients before and after the treatment, ignoring the disease evolution patterns.

In general, the manuscript is not technically sound and the data does not support their conclusions. Plus, the paper lacks the details on the methodological procedures and the statistical analysis. Worrisomely, despite using a small sample size and lacking the appropriate controls the manuscript over concludes the potential effect of adenosine on COVID-19 treatment, what may lead to a process of misinformation of the general public. Further, ethics considerations must be carefully considered since the authors demonstrate pre-existing trends before starting the research, as can be shown be passages on the text such as: “approved to receive life-saving off label treatment with inhaled adenosine”.

Therefore, I strongly recommend this manuscript to be reject for publication at Plos One.

Reviewer #4: In this short report the authors describe the use of aerosolized adenosine in a small set of COVID-19 patients hospitalised ina an ICU setting in Italy. Unfortuanetely the authors do not appear to have submitted the Table 1 that they refer to and until that is made available a substantive review is not possible.

6. PLOS authors have the option to publish the peer review history of their article (what does this mean?). If published, this will include your full peer review and any attached files.

Reviewer #1: **Yes: **Rune Andersson

Reviewer #2: **Yes: **Ali Acar

Reviewer #3: No

Reviewer #4: **Yes: **Greg Fegan

---

## [Author Response · Author response to Decision Letter 0]

13 Aug 2020

Dear Reviewers, 

wishing to thank you for your comments, we have modified the manuscrpt accordingly. A point to point reply ha been attached.

1) Please amend the title either on the online submission form or in your manuscript so that they are identical.

Reply: done as requested, we have changed the title on the online submission

2) We note that you have included the phrase “data not shown” in your manuscript. Unfortunately, this does not meet our data sharing requirements. PLOS does not permit references to inaccessible data. We require that authors provide all relevant data within the paper, Supporting Information files, or in an acceptable, public repository. Please add a citation to support this phrase or upload the data that corresponds with these findings to a stable repository (such as Figshare

or Dryad) and provide and URLs, DOIs, or accession numbers that may be used to access these data. Or, if the data are not a core part of the

research being presented in your study, we ask that you remove the phrase that refers to these data.

Reply: the sentence including "data not shown" lost at the fist check has been removed. A new reference has been included concerning the control group both in patients description in methods and results sections. All patients' informations togheter with those receiving adenosine are available at direzionesaniraria@ospedalerc.it as descriped in the point 4

3) Please include your tables as part of your main manuscript and remove the individual files. Please note that supplementary tables

(should remain/ be uploaded) as separate "Supporting Information" files 

Reply: Table 1 has been merged with the test as required.

4) Thank you for explaining the restrictions on your data. Can you please confirm whether the following proposed Data Availability

statement is accurate and suitable to appear alongside your manuscript. 

Reply: the sentence:"Data underlying the study cannot be made publicly available due to ethical concerns about sensitive patient information. The data have been

deposited on the Grand Metropolitan Hospital database and are available on qualified request (to direzionesanitaria@ospedalerc.it) according to the

UE2016/679 GDPR (General Regulation on the protection of sensitive data 2019) law", includes both experimental and control group of patients.

Regards,

Federico Bilotta

---

## [Decision Letter · Decision Letter 1]

20 Aug 2020

PONE-D-20-18007R1

Therapeutic effects of Adenosine in high flow 21% oxygen aereosol in patients with Covid19-Pneumonia.

PLOS ONE

Dear Dr. Bilotta,

Thank you for submitting your manuscript to PLOS ONE. After careful consideration, we feel that it has merit but does not fully meet PLOS ONE’s publication criteria as it currently stands. Therefore, we invite you to submit a revised version of the manuscript that addresses the points raised during the review process.

ACADEMIC EDITOR: Please see attached comments by the reviewers and provide final corrections prior to acceptance.

We look forward to receiving your revised manuscript.

Kind regards,

Muhammad Adrish

Academic Editor

PLOS ONE

Reviewers' comments:

Reviewer's Responses to Questions

**Comments to the Author**

1. If the authors have adequately addressed your comments raised in a previous round of review and you feel that this manuscript is now acceptable for publication, you may indicate that here to bypass the “Comments to the Author” section, enter your conflict of interest statement in the “Confidential to Editor” section, and submit your "Accept" recommendation.

Reviewer #1: (No Response)

Reviewer #4: All comments have been addressed

2. Is the manuscript technically sound, and do the data support the conclusions?

Reviewer #1: Yes

Reviewer #4: (No Response)

3. Has the statistical analysis been performed appropriately and rigorously? 

Reviewer #1: Yes

Reviewer #4: (No Response)

4. Have the authors made all data underlying the findings in their manuscript fully available?

Reviewer #1: Yes

Reviewer #4: (No Response)

5. Is the manuscript presented in an intelligible fashion and written in standard English?

Reviewer #1: Yes

Reviewer #4: (No Response)

6. Review Comments to the Author

Reviewer #1: In the Background I can still read that intersitial pneumonitis due to covid -19 is mostly fatal, which is no longer true and that 65-80 % of the ventilator treated patients die, which is too high in most recent clinical materials, at several centers now around 20%.

This text which I didn't found up to date is still remaining even if the authors have added a sentence telling about lower case fatality in recent patients. I recommend to delete the inactual text and base the new text on recent references only.

Reviewer #4: In the middle of the 1st paragraph on the Introduction the senetence ends with "how criticalis their ill" which I beleive would read better as "how critical their illness is." Further on in the 2nd sentence of the 1st para of the Results "Allthese" should be split into "All these". On page 20 in the in the 1st senetence under Laboratoroy response I think the "Our analysis revealed a trend to a serum decline of IL-6 levels ... " Is better put as "Our analysis revealed some limited evidence for serum decline of IL-6 levels ...".

7. PLOS authors have the option to publish the peer review history of their article (what does this mean?). If published, this will include your full peer review and any attached files.

Reviewer #1: **Yes: **Rune Andersson

Reviewer #4: **Yes: **Greg Fegan

---

## [Author Response · Author response to Decision Letter 1]

31 Aug 2020

Dear reviewers,

With the present we wish resubmit to PLOS-One, our manuscript #PONE-D-20-18007R1, entitled Therapeutic effects of Adenosine in high flow 21% oxygen aereosol in patients with Covid19-Pneumonia. We have modified the manuscript according to the few suggestions of the reviewers that we really wish to thank for their very constructive criticisms and attention. As suggested by the editor, a point to point reply has been attached and a copy of our patients’ treatment protocol has been loaded at http://journals.plos.org/plosone/s/submission-guidelines#loc-laboratory-protocols website.

Reviewer #1: In the Background I can still read that interstitial pneumonitis due to covid -19 is mostly fatal, which is no longer true and that 65-80 % of the ventilator treated patients die, which is too high in most recent clinical materials, at several centers now around 20%.

This text which I did not found up to date is still remaining even if the authors have added a sentence telling about lower case fatality in recent patients. I recommend to delete the inactual text and base the new text on recent references only.

Reply: In the present manuscript we have completely changed the sentence according to what suggested by the reviewer.

Reviewer #4: In the middle of the 1st paragraph on the Introduction the sentence ends with "how critical is their ill" which I believe would read better as "how critical their illness is." Further on in the 2nd sentence of the 1st paragraph of the Results "All these" should be split into "All these". On page 20 in the in the 1st sentence under Laboratory response I think the "Our analysis revealed a trend to a serum decline of IL-6 levels ..." Is better put as "Our analysis revealed some limited evidence for serum decline of IL-6 levels...".

Reply: We have modified the above mentioned sentences as kindly suggested by the reviewer.

Sincerely yours

Federico Bilotta

---

## [Decision Letter · Decision Letter 2]

14 Sep 2020

Therapeutic effects of Adenosine in high flow 21% oxygen aereosol in patients with Covid19-Pneumonia.

PONE-D-20-18007R2

Dear Dr. Bilotta,

We’re pleased to inform you that your manuscript has been judged scientifically suitable for publication and will be formally accepted for publication once it meets all outstanding technical requirements.

Kind regards,

Muhammad Adrish

Academic Editor

PLOS ONE

Additional Editor Comments (optional):

Reviewers' comments:

Reviewer's Responses to Questions

**Comments to the Author**

1. If the authors have adequately addressed your comments raised in a previous round of review and you feel that this manuscript is now acceptable for publication, you may indicate that here to bypass the “Comments to the Author” section, enter your conflict of interest statement in the “Confidential to Editor” section, and submit your "Accept" recommendation.

Reviewer #1: (No Response)

Reviewer #4: All comments have been addressed

2. Is the manuscript technically sound, and do the data support the conclusions?

Reviewer #1: Yes

Reviewer #4: (No Response)

3. Has the statistical analysis been performed appropriately and rigorously? 

Reviewer #1: Yes

Reviewer #4: (No Response)

4. Have the authors made all data underlying the findings in their manuscript fully available?

Reviewer #1: Yes

Reviewer #4: (No Response)

5. Is the manuscript presented in an intelligible fashion and written in standard English?

Reviewer #1: Yes

Reviewer #4: (No Response)

6. Review Comments to the Author

Reviewer #1: In Background , row 4, I suggest to change the words "mostly fatal outcome" to "risk of fatal outcome"

Reviewer #4: (No Response)

7. PLOS authors have the option to publish the peer review history of their article (what does this mean?). If published, this will include your full peer review and any attached files.

Reviewer #1: **Yes: **Rune Andersson

Reviewer #4: **Yes: **Greg Fegan

---

## [Editor Report · Acceptance letter]

30 Sep 2020

PONE-D-20-18007R2 

Therapeutic effects of Adenosine in high flow 21% oxygen aereosol in patients with Covid19-Pneumonia. 

Dear Dr. Bilotta:

I'm pleased to inform you that your manuscript has been deemed suitable for publication in PLOS ONE. Congratulations! Your manuscript is now with our production department. 

Kind regards, 

on behalf of

Dr. Muhammad Adrish 

Academic Editor

PLOS ONE